# Sample-Efficient Self-Supervised Imitation Learning

## Abstract

Imitation learning allows an agent to acquire skills or mimic behaviors by observing an expert performing a given task. While imitation learning approaches successfully replicate the observed behavior, they are limited to the trajectories generated by the expert both regarding their quality and availability. In contrast, while reinforcement learning does not need a supervised signal to learn the task, it requires a lot of computation, which can result in sub-optimal policies when we are dealing with resource constraints. For addressing those issues, we propose Reinforced Imitation Learning (RIL), a method that learns optimal policies using a very small sample of expert behavior to substantially speed up the process of reinforcement learning. RIL leverages expert trajectories to learn how to mimic behavior while also learning with its own experiences in a typical reinforcement learning fashion. A thorough set of experiments show that our method outperforms both imitation and reinforcement learning methods, providing a good compromise between sample efficiency and task performance.

## 1 Introduction

Humans have the ability to learn by observing other individuals performing certain activities. We can learn actions from individuals even without having prior information of their behavior (Rizzolatti & Sinigaglia, 2010). For example, we can learn tasks such as cooking, drawing, playing an instrument, or playing games just by watching videos. Our capabilities go beyond merely imitating; we can learn from a demonstration of a given task, despite differences in the environment, body, or objects that constitute the demonstration. Despite being studied in different areas, such as psychology (Vogt & Thomaschke, 2007; Király et al., 2013) and robotics (Schaal, 1999; Ratliff et al., 2007; Raza et al., 2012), learning by imitation recently became a prominent field in the area of artificial intelligence (AI) (Bandura & Walters, 1977; Hussein et al., 2017; Fang et al., 2019).

Reinforcement learning (RL), in turn, is one of the approaches employed in AI for learning without supervised signals, often in a trial-and-error strategy based on post-action rewards. The output of RL is a policy that specifies how an agent should act at any given time. Value-based methods compute the optimal policy by first estimating the expected value of each action using samples from the environment, and then choosing the actions with the maximum estimated expected values. However, certain environments with sparse rewards may result in a very large number of interactions of the agent to reach a reward that can be propagated to other states, making RL a strategy considerably slower than supervised learning. The assumption that the only information available to the learning agent are the immediate environmental rewards also makes the learning problem very hard (Sutton & Barto, 2018, Ch.3).

By contrast, humans can learn much faster if they have an expert showing them what to do. Imitation learning (IL) builds on this idea by mimicking the behavior of an agent (known as *the expert*) that successfully completes the task of interest, even when its behavior is not necessarily what could be consider the *optimal* behavior. However, having access to a large amount of samples provided by an expert is not the case of most application domains, in which we are often only provided with very few observations.

For addressing the disadvantages of both IL and RL, we develop a novel approach called Reinforced Imitation Learning (RIL), shown in Figure 1. Our method guides itself through the environment looking at both

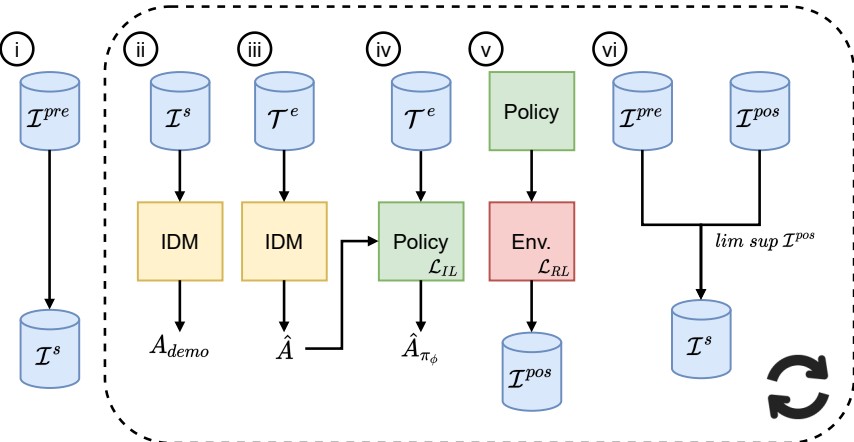

Figure 1: Reinforced Imitation Learning framework.

immediate rewards and a limited amount of observations from an expert. We show that RIL outperforms IL state-of-the-art approaches based on behavior cloning, regarding both *performance* ($\mathbb{P}$) and *Average Episodic Reward* (*AER*) metrics. We also show that RIL is competitive with (and often better than) RL approaches, though with the advantage of learning much more efficiently, being thus the strategy that presents the best trade-off between effectiveness and efficiency for real-world applications.

## 2    Related Work

We briefly review recent approaches to both reinforcement and imitation learning, starting with the former.

Deep Q-Network (DQN) (Schaul et al., 2015) is an approach that learns from the agent's experience using a deep neural network with hierarchical layers to approximate the optimal $Q$-function. Proximal Policy Optimization (PPO) (Schulman et al., 2017) combines ideas from Asynchronous Advantage Actor Critic (Mnih et al., 2016) and Trust Region Policy Optimization (Schulman et al., 2015). It uses multiple workers to avoid replay buffer and employs trust region to guarantee monotonic improvement. Like PPO, Actor-Critic using Kronecker-Factored Trust Region (ACKTR) (Wu et al., 2017) uses ideas from other methods to improve the efficiency of the learning process.

The most straightforward form of imitation learning from observation is Behavioral Cloning (BC) (Pomerleau, 1988), which treats imitation learning as a supervised problem. It uses samples comprised of the state at time $t$, the action and the resulting state $(s_t, a, s_{t+1})$ from an expert to learn how to approximate the agent's trajectory from the expert's. However, such an approach becomes costly for more complex scenarios, requiring more samples and information about the action effects over the environment. Generative Adversarial Imitation Learning (GAIL) (Ho & Ermon, 2016) solves this issue by matching the state-action frequencies from the agent to those seen in the demonstrations. GAIL uses adversarial training to discriminate state-actions from either the agent or the expert while minimizing the difference between both. It requires less expert data, though it needs substantial interactions within the environment.

Recent self-supervised approaches (Torabi et al., 2018; Gavenski et al., 2020) that learn from observations make use of the expert's transitions $(s_t, s_{t+1})$ and leverage from random transitions $(s_t, a, s_{t+1})$ in order to learn the inverse dynamics of the environment, and afterwards employ this knowledge to generate pseudo-labels for the expert's trajectories. Imitating Latent Policies from Observation (ILPO) (Edwards et al., 2019) differs from those previous studies by trying to estimate the probability of a latent action given a state. Within a limited number of environment steps, it remaps the latent actions to the corresponding actions.

There is a specific line of research in which RL uses demonstrations to assist the policy, and that could be viewed as a middle-ground (or hybrid approach) between RL and IL. DAGGER (Ross et al., 2011) iteratively produces new policies based on pulling the expert policy outside its original state space. Deep Q-learning from Demonstration (DQfD) (Hester et al., 2018) uses human demonstrations in a DQN fashion to pre-train its policy. Active Deep Q-learning with Demonstration (ARLD) (Chen et al., 2020) improves the approach of using human demonstrations to guide the learning process by introducing an active learning mechanism. Even though most of those hybrid approaches appear in the recent literature (Chen et al., 2021), note that they require the expert policy or ground-truth labels to provide feedback to the agent.

In this paper, we assume scenarios in which we do not have access to ground-truth labels for performing IL. All we have access to is the trajectory performed by an expert when acting in a given scenario, while all the rest can be learned self-supervisedly. We want to make it clear that we not have access to the actions performed by the expert. In contrast, by having access to ground-truth information, those hybrid related methods require an additional setup that our method does not, *e.g.,* training a policy to act as the expert or annotating a large number of demonstrations with the corresponding actions. Furthermore, these methods use IL as a form of divergence minimization, while our proposed approach uses IL as its primary training source and RL as a trajectory correction between observed and actual data. With that being said, in this paper we only compare our proposed method with baselines that do either reinforcement or imitation learning, but not to hybrid approaches that have access to ground-truth labels.

## 3 Reinforced Imitation Learning

Reinforced Imitation Learning (RIL) interleaves imitation and reinforcement learning steps to converge into an optimal policy in a very sample-efficient manner. RIL employs the idea of self-supervisedly learning policies based on an inverse dynamics model (Torabi et al., 2018; Monteiro et al., 2020; Gavenski et al., 2020) and then refining and improving such a policy with reward-based exploration typically performed in *q*-learning (Watkins & Dayan, 1992; Mnih et al., 2013; Kaiser et al., 2019), with the final goal of creating a new algorithm capable of using samples of an expert's trajectory to guide in the design of a policy while also using its own experiences to correct its trajectory by exploring states outside the expert's trajectories.

In this work, we base our implementation of the RL component on the original unmodified DQN architecture, since it shares interesting similarities with the self-supervised IL component (*e.g.,* both are off-policy), and we know that *exploration versus exploitation* trade-offs play a crucial role for achieving higher rewards.

### 3.1 Problem formulation

Our problem assumes an agent acting in a Markov Decision Process (MDP) represented by a five-tuple $M = \{S, A, T, r, \gamma\}$ (Sutton & Barto, 2018), in which: $S$ is the state-space, $A$ is the action space, $T$ is the transition model, $r$ is the immediate reward function, and $\gamma$ is the discount factor. Solving an MDP yields a stochastic policy $\pi(a \mid s)$ with a probability distribution over actions for an agent in state $s$ to perform.

*Imitation from observation* (Torabi et al., 2018) aims to learn the inverse dynamics $\mathcal{M}_a^{s_t, s_{t+1}} = P(a \mid s_t, s_{t+1})$ of the agent, *i.e.,* the probability distribution of each action $a$ when the agent transitions from state $s_t$ to $s_{t+1}$. In this problem, the learning agents knows neither the reward function nor the actions performed by the expert, so we want to find an imitation policy from a set of state-only demonstrations of the expert $D = \{\zeta_i\}_{i=1}^N$, where $\zeta$ is a variable-length state-only trajectory.

### 3.2 Self-Supervised Imitation Learning

Self-supervised imitation learning is a framework that usually comprises a module to learn the inverse dynamics of the environment (Inverse Dynamics Model, IDM) and a module to learn an imitation policy. The IDM is responsible for learning the actions through a transition of states $\mathcal{M}_\theta(a \mid s_t, s_{t+1})$, while the policy ($\pi_\phi$) acts as a stationary model predicting the most likely action $a$ given $s_t$. To learn these transitions, we can use $\pi_\phi$ with random weights to create a pre-demonstration dataset ($\mathcal{I}^{pre}$) comprised of ($s_t$, $a$, $s_{t+1}$)

samples. The IDM then uses $\mathcal{I}^{pre}$ to learn the inverse dynamics of the agent by finding parameters $\theta^*$ that best describe the state transitions.

Since there are no expert labels, we make use of pairs of states from expert demonstration $(s_t^e, s_{t+1}^e)$ and the IDM to predict the action responsible for all expert transitions. Subsequently, the policy model uses these self-supervised labels to learn $\pi(\hat{a} \mid s_t)$; however, considering that $\mathcal{I}^{pre}$ contains random actions, the pseudo-labels generated by the IDM can be far from the expert's. To mitigate this issue, we can rely on an iterative process, where the updated policy creates new samples $\mathcal{I}^{pos}$ and balances $\mathcal{I}^s$ with all trajectories that reach the environment goal. This process allows the model to maintain a weighted distribution between the random and updated policy samples and avoid local minima since the probability of actions vanishing in each iteration is minimal.

### 3.3 Exploration with Neural Networks and $q$-values

The second element of our proposed approach is an exploration mechanism via RL based on $q$-values and neural networks. One such a method is DQN (Mnih et al., 2013), which employs a deep neural network with hierarchical layers to approximate the optimal $Q$-function in Equation 1, where $r$ is the reward received when transitioning from state $s_t$ to $s_{t+1}$, $\alpha$ is the learning rate, $a$ is the action and $Q$ is a deep neural network.

$$Q\left(S_t, A_t\right) \leftarrow Q\left(S_t, A_t\right) + \alpha \left[R_{t+1} + \gamma \max_a Q\left(S_{t+1}, a\right) - Q\left(S_t, A_t\right)\right] \tag{1}$$

DQN implements an experience replay mechanism that stores a set of observations from the environment to update the $Q$-function with random samples. This process solves the correlation issue between sequences of observations and smooth changes in the data distribution.

For each episode, the algorithm has a probability $\epsilon$ to select a random action or use the action-value function. The value of $\epsilon$ usually decreases as training progresses to trade the exploration of early states to the exploitation of late states (near the goal). The agent executes the $\epsilon$-greedy action in the environment, which returns a reward and the next state.

### 3.4 Combining IL and RL

Reinforced Imitation Learning iteratively creates new samples using the environment and combines ideas from both IL and RL to understand how a policy can benefit from both approaches. Since the construction of a new $\mathcal{I}^{pos}$ consists of using the updated policy in the environment, where we have access to states and rewards, we introduce a reinforcement learning approach to learn by experience. Thus, the entire learning pipeline of RIL is presented in Algorithm 1, whose main steps are: (i) create dataset $\mathcal{I}^{pre}$ by using $\pi_\phi$ as $\mathcal{I}^s$ (lines 4-5); (ii) use $\mathcal{I}^s$ to learn the inverse dynamics of the environment (line 7); (iii) label the expert action $\hat{A}$ responsible for the state transitions in the expert samples $\mathcal{T}^e$ with the IDM network (line 8); (iv) use $\mathcal{T}^e$ and $\hat{A}$ to train the policy $\pi_\phi$ in an IL fashion (line 9); (v) use $\pi_\phi$ in the environment to create new state transitions $\mathcal{I}^{pos}$ and employ the temporal difference update to further learn from its own experiences (lines 10-13);

---

**Algorithm 1** Reinforced Imitation Learning

1: Initialize model $\mathcal{M}_\theta$ as a random approximator
2: Initialize policy $\pi_\phi$ with random weights
3: Generate state transitions $\mathcal{T}^e$ from expert demonstration
4: Generate $\mathcal{I}^{pre}$ using policy $\pi_\phi$
5: Set $\mathcal{I}^s = \mathcal{I}^{pre}$
6: **while** $\pi_\phi$ improves from either method **do**
7:     Update $\mathcal{M}_\theta$ by TRAIN($\mathcal{M}_\theta$, $\mathcal{I}^s$)
8:     Generate pseudo-labels $\hat{A}$ by $\mathcal{M}_\theta(\mathcal{T}^e)$
9:     Update $\pi_\phi$ by BCLoss($\mathcal{T}^e$, $\hat{A}$)
10:     **for** $e \leftarrow 1$ to $|E|$ **do**
11:         Use $\pi_\phi$ to solve environment $e$
12:         Append samples $\mathcal{I}^{pos} \leftarrow (s_t, a_t, s_{t+1})$
13:         Update $\pi_\phi$ by TDLoss($\mathcal{I}^{pos}$, $A$)
14:     $\mathcal{I}^s \leftarrow$ GOALSAMPLER($\mathcal{I}^{pos}$)

---

and (vi) use a sampling mechanism to create a new dataset $\mathcal{I}^s$ to feed the IDM network (line 14); (vii) repeat steps ii-vi until no further improvement is noticed (either when no actions change between two consecutive epochs or no significant reduction in loss is observed in consecutive epochs).

RIL uses $\epsilon$-greedy exploration in its RL-based component for learning states outside its expert. Most often, $\epsilon$-greedy approaches decrease the exploration chance as time passes, however, RIL interchanges learning from demonstration and experience. Shifting its learning approaches can result in acquiring information that might not be ideal from both perspectives. Therefore, RIL adapts its exploration behavior according to its certainty. RIL's policy uses the softmax distribution from its output to predict the action. Thus, as the policy learns to separate the different actions, it chooses actions other than the maximum *a posteriori* label less often. In every iteration from RIL, we compute the exploration ratio from the policy during the self-supervised learning component (Line 9) and use the same number for the epsilon (Lines 10-13). This approach allows the model to explore at the beginning, like $\epsilon$-greedy policies would, and as the model learns to differentiate the actions, it allows for less exploration and more exploitation. This strategy does not need to rely on a time-dependent decaying function for the exploration values (as commonly seen in RL), and allows for increased exploration when the policy finds itself in local minima.

RL approaches are not commonly designed to learn an optimal policy under few episodes. Thus, we also need to adapt the size of the experience replay: if its size is too big, there will be fewer updates during each iteration, which can lead to less desirable actions; if too small, there will be fewer samples, resulting in sub-optimal weight updates. Considering that RIL has access to the average size of each expert trajectory, we use this information to decide the size of the replay memory according to each environment. RIL sets the experience replay size to be $10\times$ the average of the expert samples it has access to for each domain.

The drawback of RIL is that part of its learning approach depends on a reward signal (RL) and another part is self-supervised (IL) in an iterative process that constantly shifts its data and labels. To overcome this issue, we modify both components in three different ways: (i) we clip the gradients from both reinforcement and imitation learning components, (ii) we add layer normalization into both IDM and policy models, and (iii) we adapt the sampling mechanism of the self-supervised strategy to reduce the inherited covariate shift. The first adaptation reduces the gradient from both learning methods. Since both parts of RIL learn with different objectives, the clipping allows for lower variance during learning. The reason for the second modification is due to the shift in data and labels from the self-supervised iterative nature that happens constantly. During the training of typical IL methods, the expert dataset contains a vast number of samples, which minimizes the consequences of the covariance shift impacting the training procedure. However, RIL contains much fewer expert samples. Thus, when the pseudo-labels from IDM significantly change from one epoch to another, a deterioration of the policy occurs. Adding layer normalization into RIL allowed for the model to learn how to correctly normalize all neurons in each layer according to the samples and allowed the policy to learn. Finally, we alter the sampling mechanism that forces the policy action distribution into the $\mathcal{I}$ given the model capability of solving the environment. Sampling from the softmax distribution allows the IDM model to rapidly learn a distribution outside $\mathcal{I}^{pre}$, which is balanced for all actions. However, this approach does not yield good results in cases with very few expert samples. The constant change from the IDM's predictions combined with the small number of examples deteriorates the policy.

To avoid this issue, we introduce an upper limit to the number of samples $\mathcal{I}$ from $\mathcal{I}^{pos}$, which we compute using Equation 2, where $n$ is a hyperparameter used to define the number of epochs it takes for the upper limit to be set to 100%, $e$ is the current epoch, and $k$ the slope of the curvature. Such an approach allows RIL to have smoother changes between epochs, reducing the covariate shift.

$$\limsup \mathcal{I}^{pos} = 1 - \frac{1}{1 + (\frac{n}{e} - 1)^{-k}}, \tag{2}$$

## 4 Experimental Methodology

Regarding the upper limit of samples, we set $k = 2$ for all main experiments in this work and give each algorithm 150 epochs ($e = 150$) for training. These values will only vary in the ablation studies we conduct in later sections. Gradient clipping values for the RL model are between $[-0.5, 0.5]$ and for the IL models between $[-1, 1]$. Details on the neural network topologies, *e.g.,* number of layers, neurons, $\alpha$, and more are all described in Section 4.1 for each environment.

We evaluate RIL and the IL related work in terms of both *Average Episodic Reward* and *Performance* metrics. $AER$ is the average reward of 100 episodes for each environment. Since $AER$ depends on an environment reward function, its value differs from task to task. $AER$ measures how well the algorithm performs the task and indicates how difficult it is for the agent to imitate the expert's behavior. *Performance* is the average reward for each run scaled from 0 to 1, where 0 is the random policy reward, and 1 is the expert. A model can achieve scores lower than zero if it performs worst than random actions and higher than one if it performs better than the expert. We do not use accuracy as a metric since achieving high accuracy in $\mathcal{I}^s$ does not guarantee solving the problems. The accuracy of the policy highly correlates with the predictions of the IDM, which does not carry information beyond the agent behavior.

For the RL approaches, we compare the sample efficiency of each method by counting how many samples each algorithm receives before reaching a certain reward, instead of computing the usual timesteps, since RIL uses expert samples as well as environment samples, allowing for a fair comparison.

We use two different DQN methods: the original version (Mnih et al., 2013) ($DQN_1$), from which we borrow several mechanisms for RIL; and its latest version (Schaul et al., 2015) ($DQN_2$), which holds the state-of-the-art for most environments experimented in this paper. We also use two other RL algorithms as baselines: PPO and ACKTR. We select these particular algorithms because they present very different approaches that end up resulting in optimal policies for the Acrobot and MountainCar environments. In this work, all IL methods apart from RIL do not employ any RL mechanism.

### 4.1 Environments and Network Topologies

The model's memory usage varies since the network topologies vary ($\leqslant 1GB$). For the IL models, both IDM and policy networks use Cross-Entropy Loss, while the Temporal Difference Loss is used by the RL model. We employ the Adam optimizer (Kingma & Ba, 2014) with its default values in all models. Below we briefly describe each environment and their respective neural network topologies and learning rates:

**i) CartPole-v1** is an environment where an agent pulls a car sideways intending to sustain a pole vertically upward as long as possible. The environment has a discrete action space composed of *left* or *right* actions, while the state space has 4 dimensions: *cart position*, *cart velocity*, *pole angle*, and *pole velocity at tip*. Barto et al. (1983) define solving CartPole as getting an average reward of 195 over 100 consecutive trials. The learning rate for this domain is $5 \times 10^{-4}$ for both models. The architecture for both IDM and policy models here is an MLP with two layers of 8 neurons activated with LReLU, and a self-attention layer with layer normalization.

**iii) Acrobot-v1** includes an agent of two joints and two links, where the joint between the two links is actuated. Initially, the links are hanging downwards and the goal is to swing the end of the lower link up to a given height. The state space consists of: $\{\cos\theta_1, \sin\theta_1, \cos\theta_2, \sin\theta_2, \theta_1, \theta_2\}$, and the action space consists of the 3 possible forces. Sutton (1996) first described Acrobot and later Geramifard et al. (2015) improved it, which is the version we use. Acrobot-v1 is an unsolved environment, *i.e.,* it does not have a specified reward threshold at which it is considered solved. The learning rate for this domain is $5 \times 10^{-5}$ for the IDM and policy models, and $5 \times 10^{-4}$ for the RL model. Both models share the same architecture, which is a two layer model with 32 neurons activated with LReLU, self-attention and layer normalization.

**ii) MountainCar-v0** environment consists of a car in a one-dimensional track positioned between two "mountains". The state-space has two dimensions, the respective car coordinates $(x, y)$, and the action space consists of 3 possible signals to move the car ($-1$, 0, or 1). To achieve the goal in this environment, the car has to acquire the required momentum and reach a flag placed on the second mountain top. Moore (1990) defines solving MountainCar as getting an average reward of $-110$ over 100 consecutive trials. Here the learning rate is set to $5 \times 10^{-3}$ for the IDM model, and $5 \times 10^{-4}$ for the policy and RL models, while the network topology is kept the same from the Acrobot-v1 environment.

**iv) LunarLander-v2** is an environment where an agent needs to land on the moon. The agent has four different actions (do nothing, move left, right and reduce the falling velocity), and the way the agent actuates influences the reward. Any movement, except for "do nothing," costs $-0.3$ reward. If it moves toward the designated landing area (always at coordinates $0, 0$), the environment returns a positive value. However,

moving away from these coordinates results in losing the previously earned reward. Finally, when reaching the floor, the environment checks whether the agent has landed or crashed and awards 100 or $-100$ points, respectively. To solve the LunarLander-v2 environment, the agent must receive a reward of 200 over 100 consecutive trials. The learning rate for this domain is $5 \times 10^{-4}$ for the IDM model, $5 \times 10^{-7}$ for the policy model, and $5 \times 10^{-6}$ for the RL model. Both models share the same architecture, which is a two layer model with 128 neurons activated with LReLU, with self-attention layers and layer normalization.

## 5 Experimental Results

### 5.1 Policy Optimization Behavior

IL and RL methods work on the same premise that an agent needs to learn an approximation to a theoretical optimal policy in the form of an MDP. Nevertheless, IL focuses on a more specific optimal policy, *i.e.,* the expert's. At the same time, RL learns how to optimize its value function, thus achieving one of many possible optimal functions for each environment. We hypothesize that RIL yields better policies than IL methods since it learns with its own experiences, while also achieving results much more efficiently than RL methods. To validate that hypothesis, we conduct an experiment where we compute the KL Divergence of a trajectory with four different policies: (i) the optimal policy ($\pi^*$), (ii) an RL policy (Mnih et al., 2013), (iii) an IL policy (Gavenski et al., 2020), and (iv) the RIL policy. Since $\pi^*$ may be one of many theoretical optimal policies, we do not use these results as a form of quantifying any of the policies created. However, upon carefully analyzing them, we can draw intuitions regarding the combination of RL and IL into a single policy.

We compute two different KL Divergence values. First, we compare all models with $\pi^*$ and compute the difference with the probability of all possible actions. This result shows how similar a policy is to the optimal one regarding its mapping of the likelihood of actions given a state. However, such difference is trivial when an agent during evaluation uses only a greedy approach for choosing an action. Thus, we also compute the KL Divergence using one-hot encodings for the specific action given a state (KL-Divergence$^*$).

Table 1 shows that RIL is the furthest policy from $\pi^*$ regarding the first metric (9.6476). This is likely to occur for two reasons: the expert may not be acting optimally; and the reinforcement learning updates can drastically alter the softmax probabilities. For the first hypothesis, we can look at Figure 2 which shows the probability of the *maximum a posteriori* action for all discretized values. When comparing $\pi^*$ to the other policies, we can see that in the middle of the *valley* $\pi^*$ has a degree of uncertainty ($\approx 33\%$ for each action), which does not show for the other policies. When comparing RL and IL, we see that the RL policy shows more similar behavior to $\pi^*$ than IL.

The IL policy carries more certainty than the other policies for all the discretized values ($\approx 58\%$). This behavior can explain the poor performance of this model. Since IL approaches only take expert samples in a supervised fashion into consideration, the model's certainty only takes into account the classification

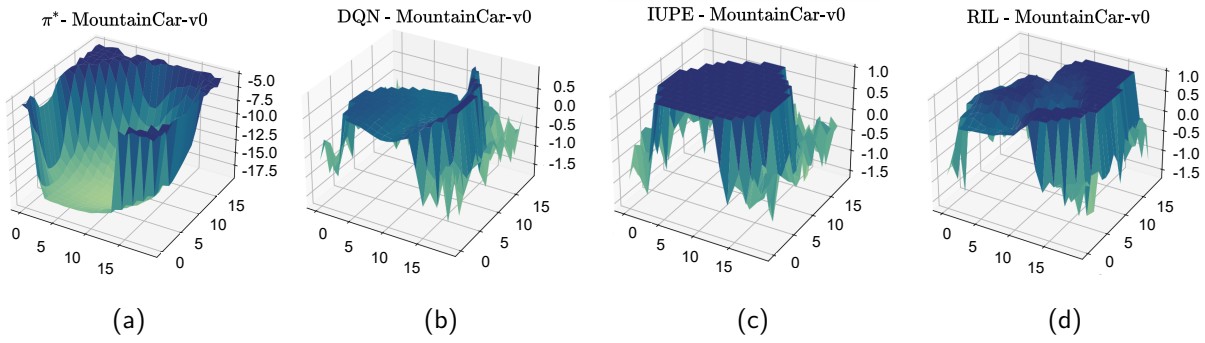

(a)          (b)          (c)          (d)

Figure 2: Visualization of the MountainCar-v0 environment. Each figure illustrates the maximum a posteriori probabilities in a 3D mesh.

Table 1: KL Divergence from all three models, when compared to an optimal policy ($\pi^*$).

| Metric | $\pi^*$ | RL (DQN) | IL (IUPE) | RIL |
|---|---|---|---|---|
| Reward | -86 | -87 | -162 | -84 |
| KL Divergence | - | 2.0869 | 4.9863 | 9.6476 |
| KL Divergence* | - | 1.7345 | 1.7345 | 0.8094 |

problem without considering the sparse reward that MountainCar presents. On the other hand, when we compare RIL to the other methods, we observe that the *valley* could be a mash of the RL and IL approaches. Although RIL still does not generate a result more in line with $\pi^*$, it balances its certainty to create a more moderate mapping for discretized values. When comparing RIL with the greedy optimal policy, our method achieves the highest similarity (0.8094). This result shows that both methods, despite their difference in probabilities, are quite similar, *i.e.,* both methods agree on the action for the same state.

Figure 3 illustrates how close RIL is from $\pi^*$ by discretizing the continuous state-space from MountainCar into a $20 \times 20$ $Q$-table, plotting the *maximum a posteriori* action and coloring the states that are equal to $\pi^*$. The figure shows that, in a discrete space, RIL is closer to the optimal policy than the remaining methods. We do not consider the reward in this episode a detrimental factor for the result in KL-Divergence*, since RL is equally distant to the optimal policy than IL. Finally, upon analyzing the regular KL Divergence, RIL should be closer to zero than RL if the reward is a significant factor for performing well in the MountainCar environment.

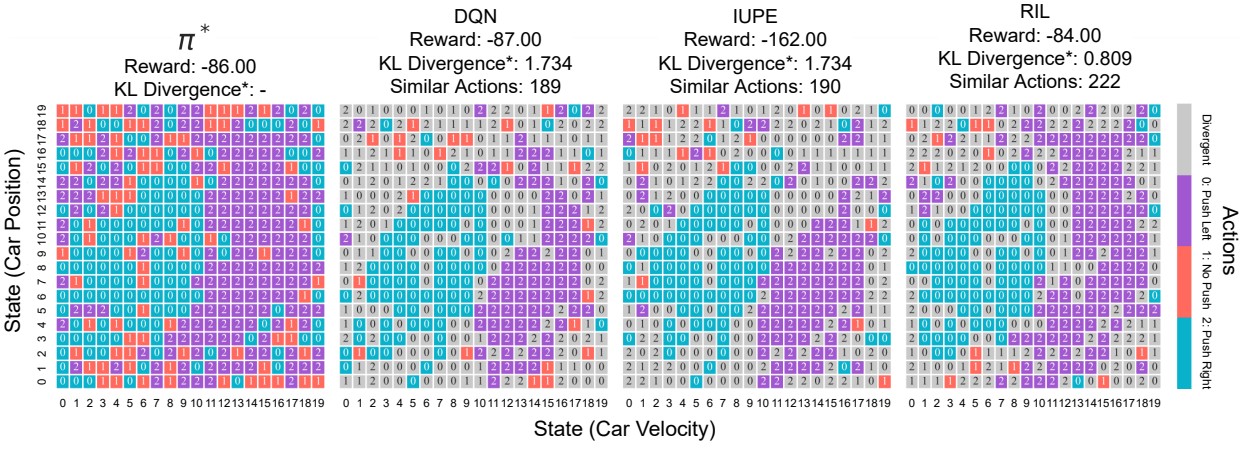

Figure 3: Comparison between policies trained in the MountainCar-v0 environment. We only color the tiles that have the same action as $\pi^*$ for easier visualization.

## 5.2 Sample Efficiency

To understand how RIL benefits from both approaches, we conduct two different experiments to validate the following research questions: (i) how each IL algorithm perform when only given one episode from its expert, and (ii) how many samples each RL method uses before solving the environment (or in the cases where the algorithm is not able to, when reaching the maximum reward).

### 5.2.1 Imitation Learning

Since RIL reaches $\mathbb{P} \geqslant 1$ with only one episode from the expert, we give the same single trajectory for all other IL methods during this experiment. The results in Table 2 show the average and standard deviation

Table 2: *Performance* ($\mathbb{P}$) and *Average Episode Reward* (AER) for each IL methods with only one expert's trajectory as data.

| Algorithms | Metric | CartPole | Acrobot | MountainCar | LunarLander | Average $\mathbb{P}$ |
|---|---|---|---|---|---|---|
| Random | AER | 18.7 | −482.6 | −200 | −182.72 | $0 \pm 0$ |
| | $\mathbb{P}$ | 0 | 0 | 0 | 0 | |
| Expert | AER | 500 | −85 | −106 | 235.96 | $1 \pm 0$ |
| | $\mathbb{P}$ | 1 | 1 | 1 | 1 | |
| BC | AER | $490.96 \pm 19.65$ | $-122.75 \pm 2.99$ | $-129.92 \pm 4.14$ | $131.84 \pm 53.25$ | $0.84 \pm 0.12$ |
| | $\mathbb{P}$ | $0.98 \pm 0.04$ | $0.91 \pm 0.01$ | $0.75 \pm 0.04$ | $0.75 \pm 0.13$ | |
| GAIL | AER | $185.07 \pm 168.25$ | $-279.02 \pm 104.91$ | $-196 \pm 10.99$ | $59.03 \pm 87.76$ | $0.36 \pm 0.23$ |
| | $\mathbb{P}$ | $0.35 \pm 0.34$ | $0.51 \pm 0.26$ | $0.04 \pm 0.12$ | $0.58 \pm 0.21$ | |
| ILPO | AER | $456.87 \pm 4.10$ | $-125.92 \pm 19.23$ | $-200 \pm 0$ | $-451.81 \pm 247.53$ | $0.29 \pm 0.75$ |
| | $\mathbb{P}$ | $0.91 \pm 0.01$ | $0.90 \pm 0.04$ | $0 \pm 0$ | $-0.64 \pm 0.59$ | |
| IUPE | AER | $144.64 \pm 11.65$ | $-232.38 \pm 50.92$ | $-198.00 \pm 6.00$ | $-203.05 \pm 35.51$ | $0.19 \pm 0.27$ |
| | $\mathbb{P}$ | $0.26 \pm 0.02$ | $0.55 \pm 0.16$ | $0.02 \pm 0.06$ | $-0.05 \pm 0.08$ | |
| RIL | AER | $\mathbf{500 \pm 0}$ | $\mathbf{-79.52 \pm 4.49}$ | $\mathbf{-100.29 \pm 1.59}$ | $\mathbf{261.73 \pm 9.91}$ | $\mathbf{1.04 \pm 0.03}$ |
| | $\mathbb{P}$ | $\mathbf{1 \pm 0}$ | $\mathbf{1.01 \pm 0.01}$ | $\mathbf{1.06 \pm 0.02}$ | $\mathbf{1.06 \pm 0.02}$ | |

of 10 different runs for each learning algorithm. We also perform an ablation study and test each algorithm with an increased number of trajectories in Section 6.3.

Considering that for the CartPole and Acrobot environments there is almost no variation in their initial states, one trajectory should be enough to achieve their goal, even though not optimally. We hypothesize that all methods have comparable results in these cases. Nevertheless, just ILPO and RIL results were good enough to achieve the goal for the CartPole environment, *i.e.,* $r \geqslant 195$. In contrast, GAIL and IUPE achieved performance around 0.30, with GAIL being only 10 reward points from the goal, though far from the expert. The Acrobot environment does not have a defined goal, but we can define that a reward close to −80 can be considered ideal, as is the case for the expert. However, only RIL was able to reach such a result. ILPO achieves a performance of 0.9, with IUPE being close with 0.8 performance points. Since a random policy achieves an AER of −482.6, both methods converge to policies closer to the expert than GAIL, which only achieves −279.02 reward points.

By contrast, MountainCar depends heavily on the agent starting position, while LunarLander alters its objective during each iteration. Having only one trajectory to learn how to mimic the expert is a significant disadvantage. This limitation is evident in the overall results for all IL methods that reach a performance of $\approx 0.06$ in MountainCar, and of $\approx -0.04$ in LunarLander. These policies are further away from the expert and the goal of the environment (−110 and 200).

Since the BC method uses the ground-truth labels, we hypothesize that this approach yield similar results to RIL, even though the number of trajectories is limited. In the CartPole and Acrobot environments, the BC method achieves results closer to the expert ($\mathbb{P} \simeq 0.9$); however, performance and rewards decrease significantly during the MountainCar and LunarLander environments. This experiment shows that RIL's capability of learning with its own experience is a substantial advantage, even with a small number of examples.

### 5.2.2 Reinforcement Learning

By comparing RIL with the IL methods, we show that it reaches better results with fewer expert samples. However, RIL uses its own experiences in the form of $q$-value mapping to create the optimal policy, which the other IL methods had no access to. Hence, we also compare it to RL methods to understand whether RIL reaches the same results with fewer samples than them. Results in Table 3 show the number of

Table 3: Average timesteps needed to reach the maximum reward (Table 4) for each algorithm in each environment.

| Environment | $DQN_1$ | $DQN_2$ | PPO | ACKTR | RIL |
|---|---|---|---|---|---|
| CartPole | 211,000 | 64,500 | 15,000 | 169,500 | **14,800** |
| Acrobot | 427,500 | 498,000 | 98,000 | 482,000 | **16,365** |
| MountainCar | 224,500 | 155,500 | 680,000 | 507,500 | **29,745** |
| LunarLander | 357,000 | **239,000** | $154,000^{\dagger}$ | 646,000 | 281,204 |

timesteps needed for reaching each method's maximum reward. Since RIL uses off and online learning, we compute both the samples used during the RL training and the expert samples used during the IL training in these results. Thus, if the expert trajectory has a size of 500 samples, for each RIL's iteration we count the number of timesteps from the RL component training plus 500.

As expected, $DQN_1$ and $DQN_2$ yield similar results reaching their maximum reward, though $DQN_2$ solves most environments while $DQN_1$ does not. $DQN_2$ achieves its maximum reward more efficiently than the other RL methods, besides achieving higher or comparable rewards. $DQN_2$ presents $\mathbb{P} = 0.92$, while PPO and ACKTR achieve $\mathbb{P} \approx 0.85$, with the difference between PPO and $DQN_2$ being negligible for CartPole and significant for MountainCar. The exception for $DQN_2$ is the Acrobot environment where PPO's use of a smoother exploration mechanism can achieve better results with fewer samples $(524, 500)$. We observe the same behavior with RIL. Since the $\epsilon$-greedy strategy used during the RL training initiates with a value corresponding to the exploration rate from the policy in the previous epoch, the exploration becomes less frequent. Thus, it allows for a more efficient path for reaching its maximum reward. While ACKTR achieves the best result among the RL methods for the MountainCar, it also requires $\approx 500,000$ timesteps $(477, 755$ more than RIL). We note that for MountainCar, the number of timesteps PPO needs to reach its maximum reward is lower than all the other algorithms. Since PPO does not reach the environmental goal, we do not consider it a relevant result.

These experiments show that the trajectory of the expert can be used as a shortcut, allowing RIL to achieve its maximum reward and the goal of all environments more efficiently than the RL approaches. Nevertheless, RIL inherits the problem from IL of using an expert's trajectory as a guide, making it quite difficult to learn how to behave in those environments in which the goal constantly shifts, *e.g.,* LunarLander.

## 5.3 Quantitative Results

Table 4 presents all results for both paradigms. In this experiment, we compare the capability of each method to learn a policy capable of reaching the highest reward, and we also show the average $\mathbb{P}$ when compared to the expert performance. Considering that we previously hinder all IL methods with a single expert trajectory, we use 100 different trajectories $(\approx 100 \times$ more samples). For the RL approaches, we train each algorithm for $2,000,000$ timesteps.

As expected, $DQN_1$ achieves lower rewards than every other RL algorithm, in agreement with our premise that the RL component of RIL itself should not be enough to solve the environment. Except for the CartPole environment, in which all methods reach the $r \geqslant 195$, the first DQN was closer to the random policy than the expert reward. By comparison, $DQN_2$ achieves the environmental goal, and the Acrobot ideal reward, for almost all environments but MountainCar, something that no other RL approach was capable of. When analyzing PPO's results, we see that even though it achieves the best result over all other RL methods for Acrobot, it performs worse than ACKTR and $DQN_2$ for the other environments, while ACKTR achieves the best result for MountainCar while being worse in the rest of the environments. We expect MountainCar to be a complex task considering it has a sparse reward function and a goal that rewards less exploration from the policy. On the other hand, for the LunarLander environment, even though most of the RL algorithms did not achieve the goal, *i.e.,* $r \geqslant 200$, we note that achieving a positive result can be quite difficult. Since its goal and reward system do not have a strong correlation, such as in CartPole, and its goal is not fixed, unlike

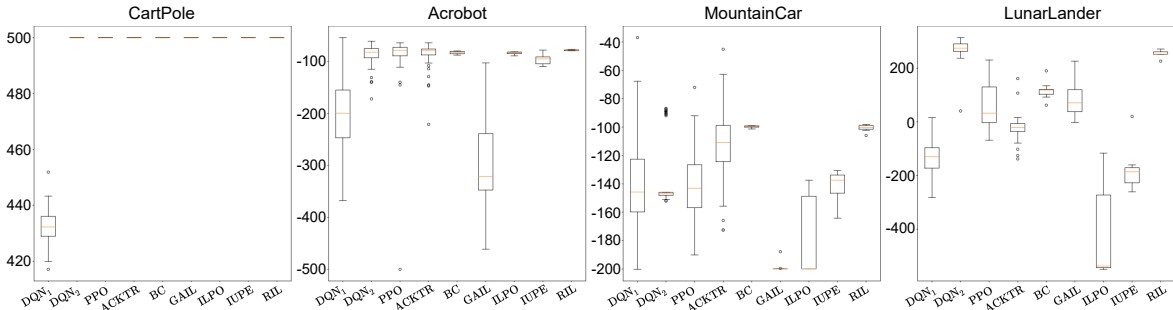

Figure 4: Boxplot of the *Average Episodic Reward* for all methods and environments.

*e.g.,* MountainCar and Acrobot, LunarLander has the lowest performance among all tested environments. The average $\mathbb{P}$ for all RL approaches is approximately 0.8.

When comparing the IL methods (except for BC), it becomes clear that they can have significant difficulties when dealing with the LunarLander environment as well. While the RL methods achieve a positive result, the IL strategies deteriorate over time. We assume that such behavior originates from the policies learning how to mimic the expert's landing position, which does not correlate to the goal. Since these methods lack the reward signal to correct themselves, the result is closer to the random policy than the expert.

We observe that the IL methods achieve, on average, a higher result in the Acrobot environment ($\approx 81.69$) than the RL methods ($\approx 86.87$ – excluding $DQN_1$). This is due to the fact that IL methods learn an optimal trajectory without much exploring. However, the IL approaches tend to replicate the average of the actions in a given state as the number of trajectories grows. This is a good thing in cases such as CartPole and Acrobot because the expert's states do not vary as much. The policies can predict the correct answers even when a particular state was absent in the learned trajectories or rapidly correct itself. Given the self-supervised nature from these methods, we observe in MountainCar a decrease in reward. This outcome happens because those algorithms make use of pseudo-labels and only approximate the expert's action. An incorrect action might cause the car to lose momentum in this environment, resulting in fewer reward points. A solution would be to use the ground-truth labels from the expert.

Table 4: Quantitative results for all RL and IL algorithms used in this work as baselines. We also display the average *performance* of all environments. $DQN^1$ is the unmodified DQN architecture Mnih et al. (2013), while $DQN^2$ is the version from Schaul *et al.* Schaul et al. (2015).

| Environments | Reinforcement Learning | | | | Imitation Learning | | | | |
| --- | --- | --- | --- | --- | --- | --- | --- | --- | --- |
| | $DQN_1$ | $DQN_2$ | PPO | ACKTR | BC | GAIL | ILPO | IUPE | RIL |
| CartPole | 431.87 ±5.31 | **500.00** **±0.00** | **500.00** **±0.00** | 487.70 ±64.76 | **500.00** **±0.00** | **500.00** **±0.00** | **500.00** **±0.00** | **500.00** **±0.00** | **500.00** **±0.00** |
| Acrobot | -191.51 ±64.29 | -87.83 ±27.96 | -83.43 ±23.29 | -89.36 ±24.89 | -82.92 ±2.63 | -83.12 ±20.95 | -83.84 ±2.50 | -78.10 ±10.56 | **-75.72** **±4.49** |
| MountainCar | -145.00 ±31.18 | -135.28 ±24.10 | -142.16 ±21.67 | -112.55 ±21.19 | **-99.69** **±0.69** | -186.74 ±0.65 | -177.56 ±27.77 | -130.70 ±15.23 | **-100.37** **±2.50** |
| LunarLander | -127.64 ±70.58 | **273.07** **±37.92** | 105.24 ±51.90 | 85.85 ±64.72 | 214.93 ±5.56 | 83.56 ±65.26 | -421.62 ±180.41 | -211.2 ±40.77 | **266.55** **±19.34** |
| Average $\mathbb{P}$ | 0.53 | 0.92 | 0.83 | 0.88 | 1.01 | 0.70 | 0.42 | 0.67 | **1.04** |

BC has results similar to the best RL method ($\text{DQN}_2$) and close to RIL with 1.01 performance points. While the performance is similar to the expert, it has access to ground-truth labels, which can be hard to acquire or ineffective if an agent has to learn the environment to produce a significant number of annotated trajectories. Hence, RIL's overall performance is a significant improvement over the current IL methods. RIL performs equal or greater than the expert on all environments without any ground-truth labels from its expert. It achieves higher rewards than the RL methods as well, with the single exception of $\text{DQN}_2$ on the LunarLander environment. For that, we analyze the standard deviation from both methods and the boxplot graph presented in Figure 4. RIL achieves a reward of 266.55 with a standard deviation of 19.34, while $\text{DQN}_2$ achieves 273.07 reward points with a higher deviation (37.92). RIL stands within the interval from $\text{DQN}_2$ and achieves a reward far higher than needed for solving the LunarLander environment, *i.e.*, $r \geqslant 200$. We hypothesize that applying a harsher gradient clipping during the RL training within RIL is responsible for that result. A solution would be to decrease the values from the IL training while increasing for the RL component as the epochs progress.

We note that the standard deviation from RIL is smaller than every other method but BC. The average deviation for all environments is $\approx 6.58$, while $\text{DQN}_2$, which has the best results among the RL methods, is $\approx 23$. For BC, the average standard deviation was 2.22, only 4.36 lower than RIL, a difference that is not really significant considering that BC uses ground-truth labels for the policy to learn. For that reason, we plot the interval for all methods in all environments in Figure 4, allowing us to understand how RIL compares to other methods in terms of variance (stability).

Apart from BC, we observe that RIL presents the lowest variance among all methods. In the case of MountainCar, RIL does not achieve the highest possible value – ACKTR is the method that achieves it, with a non-outlier maximum value of $\approx -60$, though RIL's behavior surpasses the median behavior of ACKTR. Note that RIL presents a very low variance, which translates into stability, a desired property for a policy to perform well when in production settings.

A similar thing happens in the LunarLander environment. Even though $\text{DQN}_2$ has the highest average of 273.07, and maximum value of $\approx 290$, RIL's behavior is within the interval of the RL method, with a difference of 6.52. We recall that LunarLander is very difficult for the IL methods because the landing position strongly correlates to the final reward. We are computing the average of the environment over 100 episodes, *i.e.*, 100 different landing positions. No other environment we use in this work has the same characteristic of a moving goal.

RIL's behavior shows that employing a hybrid RL/IL approach results in policies with lower variance. Moreover, by comparing its stability with the RL approaches, we observe that the adaptation capability of the latter is not as on-point as RIL's. We observe the same trend for the IL approaches, though their average result is more in line with RIL's in the Acrobot environment.

## 6 Discussion

### 6.1 Off-policy and Imitation Learning

When trying to learn an optimal policy, all learning methods need to find a good trade-off between instant (greedy) locally-optimal behavior and sub-optimal (though perhaps better in the long term) behavior, giving rise to the well-known *exploration vs exploitation* dichotomy. Off-policy methods make use of two different policies to address this trade-off. The first one is the target/optimal policy, while the second is the behavior/exploration policy. The behavior policy uses exploration mechanisms to generate state transitions, which will be used by the target policy during learning. Off-policy methods have access to its behavior policy, its states transitions, and output in the form of $(s_t, s_{t+1}, \pi(A_t \mid s_t), r)$. Such methods can use this information paired with an importance sampling mechanism to estimate expected values under a distribution.

On the other hand, imitation learning addresses the trade-off by leveraging an unknown expert policy. The most simplistic approach, Behavioral Cloning, uses the state transitions and actions $(s_t, s_{t+1}, a_t)$ in a supervised manner to learn the expert's optimal behavior without the need for exploration. However, this becomes costly as the domain's complexity rises due to the need for additional data. Other IL approaches

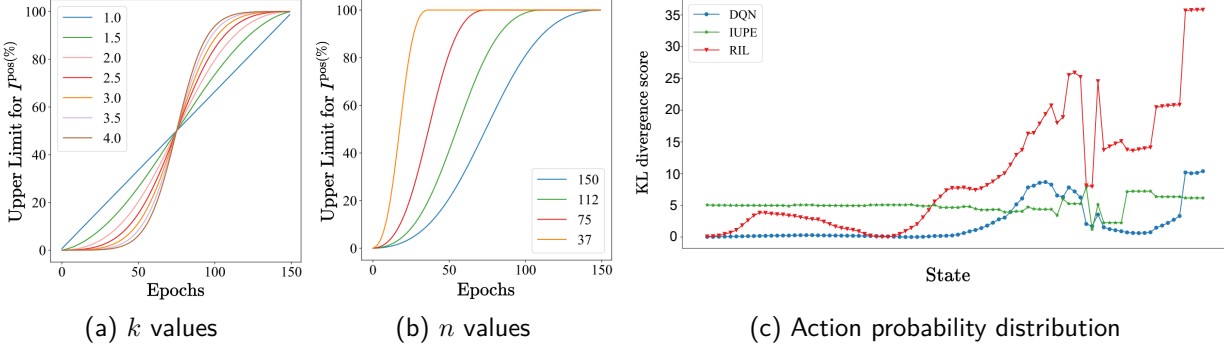

(a) $k$ values        (b) $n$ values        (c) Action probability distribution

Figure 5: Figures 5a and 5b present different values for $k$ and $n$ and their effect in Equation 2, while Figure 5c shows the action probability distribution representation of $\pi^*$ for each algorithm in a given trajectory within the MountainCar-v0 environment.

often only have access to the state transitions $(s_t, s_{t+1})$ since state transitions are more accessible than entire annotated datasets. In this case, the IL policies cannot use importance sampling to learn the optimal solution for given a problem. The action performed by the expert is not accessible by the policy either. Hence, an iterative process such as the one in RIL is vital to improve the policy. By using pseudo-labels that constantly change due to weight updates from the IDM, the policy receives different $\hat{a}$ for the same $s_t$. This behavior allows the weights from the policy to receive more updates, which helps at avoiding local minima. We illustrate an example of this behavior in Figure 5c, where the changing of probabilities becomes more abrupt for the policy. For that reason, the IL component in RIL cannot be considered as an off-policy method, nor the RL component as an IL method. We borrow mechanisms from both IL and RL, which are vital for the performance of RIL.

## 6.2 Iterative vs Sequential Learning

RIL combines RL and IL components following the insight that both offline and online learning can provide benefits in terms of both efficiency and effectiveness. In this section, we compare two possibilities regarding the use of the IL and RL components within our framework RIL.

In the regular setting, IL and RL are intertwined within the same iterative learning process, and thus the reward signal helps correcting the policy path by visiting unexplored states and approximating the policy from the expert's trajectories. In an alternative setting, we execute the IL component first and afterwards train the RL component to improve the policy with its own experiences. For that, we run the IL component for 100 epochs and afterwards the RL for $2,000,000$ timesteps. Results for this setting are presented in Table 5 denoted as RIL*.

Table 5: *AER* and *Performance* ($\mathbb{P}$) for all environments with sequenced (RIL*) and iterative (RIL) approaches.

| Environment | RIL* | RIL |
|---|---|---|
| CartPole | $382.95 \pm 208.29$ | $\mathbf{500.00 \pm 0.00}$ |
| Acrobot | $-493.63 \pm 44.61$ | $\mathbf{-75.72 \pm 4.49}$ |
| MountainCar | $-200.00 \pm 0.00$ | $\mathbf{-100.29 \pm 1.59}$ |
| LunarLander | $-114.09 \pm 56.84$ | $\mathbf{266.55 \pm 19.34}$ |
| Average $\mathbb{P}$ | $0.22$ | $1.04$ |

During this investigation, we observe that two possible scenarios can occur when using the methods in sequence: (i) if we keep a low number of expert's trajectories, the IL policy can get stuck in bad local minima, from which the RL policy cannot escape from; or (ii) the IL policy comes close to solving the environment, but due to the $\epsilon$-greedy exploration nature of the RL policy, all the learned behavior is lost in the early exploration steps. The IL component stuck on bad local minima occurs in MountainCar, where RIL* achieves $r = -200.00$ (the minimum environment reward). RL deteriorating the IL behavior scenario occurs in CartPole and Acrobot, where RIL* achieves $382.95$ and $-493.63$, respectively. We confirm that consecutively swapping between both approaches indeed help the policy to adjust itself better. By applying the IL component, the policy can correct the exploring mistakes with the expert's trajectory, whereas applying the RL component allows the policy to deviate from the expert whenever needed.

### 6.3 Impact of the amount of expert samples

In the previous experiments, we showed results with 1 and 100 expert trajectories. While the difference seem to be minor, we want to understand how different numbers of trajectories impact RIL. Table 6 presents results when using 1, 25, 50, 75, and 100 different expert's trajectories.

Given the forgiving nature of the CartPole environment, we expected no differences when using more (or fewer) trajectories. RIL achieves 500 with no standard deviation between runs in the environment. The Acrobot environment presents low variance among the number of trajectories. Using 100 different trajectories, RIL achieves the best reward ($-75.72$). However, this improvement represents a performance of 1.02, less than 1% from using a single trajectory. Considering the cost of producing 100 trajectories, we assume that using a single trajectory for this environment is enough. For the MountainCar environment, we observe that all results stay within the standard deviation from both Tables 2 and 4. We hypothesize that RIL can correctly balance the experiences from its IL and RL components and access the proper action given a state. This behavior is crucial given that IL methods perform worse than RL in this environment. The policy's own experiences can over-correct the trajectory to predict better actions and prevent the agent from slowing

Table 6: RIL performance when using a variable amount of trajectories from the expert.

| Environment | Trajectories | Samples Amount | Reward |
|---|---|---|---|
| CartPole | 1 | 500 | 500.00 |
| | 25 | 12,500 | 500.00 |
| | 50 | 25,000 | 500.00 |
| | 75 | 37,500 | 500.00 |
| | 100 | 50,000 | 500.00 |
| Acrobot | 1 | 74 | -79.52 |
| | 25 | 1,946 | -77.90 |
| | 50 | 4,089 | -77.06 |
| | 75 | 6,163 | -76.50 |
| | 100 | 8,099 | **-75.72** |
| MountainCar | 1 | 106 | -100.29 |
| | 25 | 2,466 | -100.20 |
| | 50 | 4,974 | -101.90 |
| | 75 | 7,512 | **-99.78** |
| | 100 | 10,046 | -100.37 |
| LunarLander | 1 | 453 | 261.73 |
| | 25 | 10,314 | 221.80 |
| | 50 | 20,846 | 246.32 |
| | 75 | 32,311 | 264.20 |
| | 100 | 42,384 | **266.55** |

the car's momentum. In the LunarLander environment, we note that as the expert samples increase, the policy tends to deteriorate. We attribute such a behavior to the IL component, because as the number of expert trajectories grows, more data is available to the policy during the behavioral cloning update. These weight updates deteriorate the policy since the expert's positions do not align with the goal in all episodes. Thus, when reaching 100 trajectories, the policy has far more examples to generalize and perform well. As is the case for all other environments, the RL component helps the policy to maintain a higher reward.

These results show that even though RIL benefits from a more diverse expert dataset, the overall gain is usually not significant to justify the cost of acquiring more extensive datasets. When paired with fewer samples, the IL component helps the policy mapping the actions and guiding the agent to an early trajectory. In contrast, the RL component helps correcting the trajectory by maximizing the reward signal.

### 6.4 Upper limit from $\mathcal{I}^{pos}$

In this section, we show how different values for $k$ and $n$ from Equation 2 affect the performance of RIL. We first investigate hyperparameter $k$ to understand how a more relaxed upper limit impacts RIL. Next, we test different values for $n$, which controls the shift from $\mathcal{I}^{pre}$ and $\mathcal{I}^{pos}$ to $\mathcal{I}^s$, and how different values of this hyperparameter can affect RIL.

#### 6.4.1 Varying $k$

Hyperparameter $k$ allows us to understand whether RIL benefits from a more strict upper limit or a more relaxed one. A more rigid upper limit results in fewer shifts in the labeled data, reducing the covariance shift between iterations. In contrast, a more relaxed upper limit allows $\mathcal{I}^s$ to receive a larger number of $\mathcal{I}^{pos}$, becoming closer to the expert trajectories and constantly shifting data for the IDM. Figure 5a presents the upper limit for $\mathcal{I}^{pos}$ by varying $k$ from 1 to 4. When $k = 1$, the upper limit grows linearly with the epochs, and as $k$ increases, the growth becomes exponential in the initial epochs and logarithmic in the final epochs. Table 7 shows the AER for a policy trained with different $k$ values (we keep the other hyperparameters the same: $n = 150$, $|\mathcal{T}^e| = 1$). For CartPole, the value of $k$ makes no difference to RIL results. Since the environment can be easily solved, we had not expected any differences for this hyperparameter. However, as we analyze the other environments, it becomes clear that it is critical to guarantee smoother transitions between transition samples for a proper policy. In Acrobot, as $k$ increases, the policy degrades. We believe that is because the random policy's transitions are not enough for the IDM to create pseudo-labels for the expert's trajectory properly. Hence, as the values decrease, the policy achieves higher rewards, with the optimal value being 2. As for MountainCar, we observe that just as in Table 6, the results do not show a significant variation. Except for the values of 1 and 4, which yield rewards of $\approx -150$, we observe a standard deviation of 5.08, which is lower than all RL learning approaches in Table 4. In LunarLander, we see a rapid deterioration of the reward as $k$ deviates from 2. We observe the same behavior in MountainCar, where 1 provides the worst result and 4 the second worst result. We attribute such a behavior to RIL's capability of using its RL component to counteract the IL sub-optimal knowledge, thus acquiring higher rewards than the RL methods by themselves and comparable results among different $k$ values. These results show that using $k = 2$ is the best strategy for RIL. It offers a good trade-off by maintaining $\mathcal{I}^s$ mainly from $\mathcal{I}^{pre}$ earlier in

Table 7: *Average Episode Reward* (AER) for varying $k$ values with $n = 150$ and $|\mathcal{T}^e| = 1$.

| Environments | $k$ | | | | | | |
| --- | --- | --- | --- | --- | --- | --- | --- |
| | 1 | 1.5 | 2 | 2.5 | 3 | 3.5 | 4 |
| CartPole | 500.00 | 500.00 | 500.00 | 500.00 | 500.00 | 500.00 | 500.00 |
| Acrobot | -81.84 | -79.95 | **-79.52** | -80.37 | -81.05 | -83.69 | -85.43 |
| MountainCar | -150.3 | -102.30 | **-100.29** | -105.30 | -109.20 | -112.80 | -149.90 |
| LunarLander | 132.00 | 145.36 | **266.55** | 247.50 | 209.40 | 174.6 | 171.00 |

the process, while later primarily from $\mathcal{I}^{pos}$. This behavior allows the IDM to learn the action transitions from $\mathcal{I}^s$ without suffering from heavy shifts in terms of dataset.

### 6.4.2 Varying $n$

Hyperparameter $n$ alters how early in the learning process the upper limit will reach its maximum value of 1 (or 100%). Figure 5b presents the behavior of the upper limit for 150 epochs with $n$ varying in $\{37, 75, 112, 150\}$, which is equivalent to 25%, 50%, 75%, and 100% of the epochs. Table 8 shows the AER for a policy trained with distinct $n$ values, while all other hyperparameters stay the same, *i.e.,* $k = 2$ and $|\mathcal{T}^e| = 1$. The policy once again achieves the maximum reward for the CartPole environment independently of its hyperparameters. On the other hand, as $n$ decreases the policy deteriorates in the remaining environments. We believe this is due to the high variance in samples for $\mathcal{I}^s$. As the upper limit curvature becomes steeper, the faster $\mathcal{I}^s$ becomes $\mathcal{I}^{pos}$, *i.e.,* fewer samples from $\mathcal{I}^{pre}$ are used to complement $\mathcal{I}^s$. This shift in data can deteriorate the IDM capability of predicting the correct action given $(s_t,\ s_{t+1})$. However, the results achieved by the policy are still solid enough to solve each environment. They corroborate our hypothesis that the RL component in RIL can partially use the expert's knowledge to achieve good results with its own experiences. These results show that using $n = 150$ (total number of epochs) is the best strategy for RIL.

Table 8: Average Episode Reward (AER) for distinct $n$ values.

| Environments | $n$ | | | |
| --- | --- | --- | --- | --- |
| | 150 | 115 | 75 | 37 |
| CartPole | 500.00 | 500.00 | 500.00 | 500.00 |
| Acrobot | **-79.52** | -80.90 | -82.23 | -85.31 |
| MountainCar | **-100.29** | -100.50 | -106.70 | -112.10 |
| LunarLander | **266.55** | 163.70 | 148.10 | 120.98 |

## 7 Usage of sub-optimal experts

As a premise for IL methods, the expert acts as an optimal policy in the environment. Thus, we experiment on how RIL acts when given sub-optimal experts. We hypothesize that with the use of an increasingly degrading expert, RIL's performance should degrade accordingly. Nevertheless, considering its capability of using its own experiences, the drop in performance should not be drastic, at least in theory. We do not use the LunarLander environment in this experiment, considering its aforementioned complexity (high correlation with landing position, which is variable). For the other environments, we generate experts by continuously reducing their performance in solving the task. Table 9 shows the reward from the expert that was employed for learning RIL's policy and $\pi_\phi$ results for all environments.

The first value for each environment is the expert used in the main experiments. Each expert then becomes gradually worse (decreasing values of reward). The CartPole policy achieves the maximum reward even with an expert policy distant from its goal, *e.g.,* $r \geqslant 195$. This behavior was verified in every other experiment, and points to the simplicity of solving this particular environment. The same behavior does not occur for Acrobot and MountainCar, which followed our hypothesis of gradual degradation. The degradation of the expert's policy results in worse $\pi_\phi$, though note that such a degradation is sub-linear with the decrease in reward. Even with severely sub-optimal experts, RIL is capable of achieving rewards equal or very close to the environment's goal. As the expert decayed 50 reward points for Acrobot, $\pi_\phi$ decayed only $\approx 0.80$. The same happens to MountainCar, where the expert decay is 44 reward points, while $\pi_\phi$ decreases only 10.91 points. The results from MountainCar provide evidence that the trajectory of the expert helps the policy in learning how to act in an environment while the RL component helps adjusting that behavior towards the maximum reward.

Table 9: *Average Episodic Reward* for the learned policy given sub-optimal trajectories from an expert.

| Environment | Expert Reward | Policy Reward |
|---|---|---|
| CartPole | 500 | 500.00 |
| | 400 | 500.00 |
| | 300 | 500.00 |
| | 200 | 500.00 |
| | 100 | 500.00 |
| Acrobot | -85 | **-79.52** |
| | -100 | -81.41 |
| | -150 | -80.27 |
| | -200 | -80.03 |
| | -250 | -82.72 |
| MountainCar | -106 | **-100.29** |
| | -140 | -104.07 |
| | -150 | -111.20 |

## 8 Conclusions and Future Work

In this work, we proposed Reinforced Imitation Learning (RIL), a framework that combines IL and RL components into an intertwined iterative process. RIL uses unlabeled expert samples and its own experiences to achieve state-of-the-art results in distinct benchmarking environments. The IL component of RIL uses unlabeled expert trajectories to guide the policy into a theoretical optimal policy. The RL component, in turn, can adjust the policy towards a more coherent path by exploring the $q$-value functions from its own experiences.

RIL offers two main advantages: (i) it is capable of working even with very few expert's trajectories due to its self-supervised learning strategy; and (ii) it achieves state-of-the-art results without large amounts of data or timesteps due to its capability of leveraging from the advantages of both IL and RL paradigms. Compared to other IL methods, RIL achieves better results with fewer samples while also matching their performance in scenarios with a high number of expert trajectories. Experiments also show that RIL achieves comparable (and often better) results than RL baselines though in a much more efficient way (*i.e.,* fewer timesteps).

As future work, we intend to adapt RIL to continuous environments, and also to scenarios in which the states are represented by visual information. Since the mechanisms implemented in RIL require discretization of continuous environments, we intend to perform modifications that will allow RIL to perform continuous exploration seamlessly. Environments whose states are represented as images (visual domains) such as Atari games are often harder to learn due to their larger state-spaces, so it will be interesting to verify whether RIL can also achieve state-of-the-art results in those scenarios.

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
