# OpenReview forum: "Sample-Efficient Self-Supervised Imitation Learning"
_TMLR — Rejected by TMLR_

### Review · Reviewer_3Jg7 · 2022-05-28

**Summary Of Contributions:**

_tldr: BCO + DQN._

This paper proposes an RL algorithm for the setting where the agent is also given access to a set of state-only expert demonstrations. The proposed method alternates between (1) running behavioral cloning from observations (Torabi '18), which performs imitation learning on the expert demos; and (2) deep Q-learning. On 4 simple tasks, the proposed method outperforms baselines that only perform imitation learning (i.e., omit step 2) and baselines that only perform RL (i.e., omit step 1).

**Broader Impact Concerns:**

N/A. The proposed RL algorithm doesn't directly raise any concerns in this regard.

**Requested Changes:**

**[very important]** Compare to recent, competitive baselines that make the same assumptions as the proposed method. Use more challenging tasks, like those used in these recent works (e.g., the Deepmind Control Suite, CARLA).

**[very important]** Some parts of the writing were imprecise (or worse, wrong); many claims were missing citations. I would highly recommend revising the paper (especially the abstract and Sections 1 -- 3) to make sure that the writing is precise. A few examples:
* "[imitation learning approaches] are limited to trajectories where ..." I didn't understand what this means.
* "reinforcement learning does not need a supervision signal" -- I don't think this is true: the reward is a supervision signal.
* "a method that learns optimal policies" -- The paper doesn't include any analysis proving that the proposed method always returns the optimal policy; "optimal" isn't formally defined.

**[medium importance]** The paper seems much too long. I'm not sure what the general standard for TMLR is, but it seems like the paper could be made 30-50% shorter without losing anything important. For example, section on ablation experiments could be shortened to 1 page.

**[medium importance]** "by counting how many samples..." This seems to hide an important difference: expert samples and agent samples don't necessarily cost the same amount. I would highly recommend comparing to baselines that make the same assumptions. In addition, it could be interesting to look at a sort of Pareto frontier of performance after (say) 10k samples vs % of expert samples.

**[less important]** Minor writing comments
* "recently became ... [Bandura '77]" -- It's a bit odd to refer to a 44 year old paper as recent.
* "post-action rewards" -- Unclear what this means.
* "Humans can learn much faster" -- Cite.
* "not necessarily what could" -- Cite.
* In the related work section, I'd recommend focusing on just the most relevant prior methods (those that combine RL + IL).
* "we do not have access to ground truth labels .... we have a trajectory performed by an expert" -- This seems like a contradiction.
* "very sample-efficient" -> "sample efficient"
* "q-learning" -> "Q-learning"
* "RIL employs ... outside the expert's trajectories" -- Potential run-on sentence.
* "DQN ... shares similarities with ... IL" -- I didn't understand this claim. I would recommend explaining it.
* "Markov Decision Process" -> "Markov decision process"
* "pseudo-labels ... can be far from the experts" -- Why?
* Sec. 3.3 -- I found this section confusing because DQN is an RL algorithm, not an exploration algorithm. I'd recommend explaining that the method is an RL+IL method, and mentioning that the RL algorithm is discussed in Sec. 3.3 and the IL algorithm is discussed in Sec. X.
* Alg 1, L10: "e" was defined to indicate things coming from the expert, so it's unclear how one can enumerate over e.
* "softmax distribution" -- I'd recommend discussing the similarities/differences with Boltzmann exploration.
* Sec. 4.1 could be moved to the appendix.

**Strengths And Weaknesses:**

**Strengths**
* The paper includes fairly detailed ablation experiments.
* The high-level problem studied in this paper, improving the sample efficiency of RL methods, is very important.

**Weaknesses**
* The paper is hard to follow. After reading the introduction, I was unsure exactly what problem the paper was solving (e.g., would the method be evaluated based on reward maximization or divergence minimization).
* The proposed method makes more assumptions than the baselines. Some baselines don't use the reward signal, the rest of the baselines don't use the expert demos. This makes the comparisons seem a bit unfair.
* The experiments are only on very simple tasks.

---

> ### Author Response · Authors · 2022-06-07
> **Comments to Reviewer 3Jg7**
>
> Thank you for your input on our work. We've read your points and agree on changing the paper to better explain its objective and what we are calling an optimal policy.
>
> We need to point out that RIL only works with discrete environments, as is the case of [1, 2, 3], and hence we are not be able to test our algorithm in environments with continuous actions, e.g., CARLA and Deepmind Control Suite.
>
> We've recently found an environment for autonomous driving ([Bark Behavior Benchmark](https://github.com/bark-simulator/bark)) with a discrete set of actions, and we are finishing the set of experiments there. Would that suffice? Are there any other discrete environments you would like to suggest?
>
> [1] Provable Representation Learning for Imitation with Contrastive Fourier Features, NeurIPS 2021
>
> [2] MICo: Improved representations via sampling-based state similarity for Markov decision processes, NeurIPS 2021
>
> [3] Adapting to Reward Progressivity via Spectral Reinforcement Learning, ICLR 2021

---

> > ### Comment · Reviewer_3Jg7 · 2022-06-08
> > **Response**
> >
> > Thanks for clarifying that the method only works with discrete actions. But, this should be a pretty easy limitation to lift, right? I.e., it'd be straightforward to do BCQ + TD3 instead of BCQ + DQN. This would allow a more straightforward comparison with prior methods.
> >
> > Or, perhaps it would make sense to compare on Atari. There's been some work looking at using demos + RL on Atari [1], which could serve as baselines.
> >
> > At a high-level, my concern with the original version of the paper was that the proposed method was making more assumptions that the prior methods. So, it seems like the onus is to show either that (1) the proposed method also outperforms competitive baselines that use the exact same assumptions, or (2) that we get a huge performance boost from a relatively minor additional assumption.
> >
> > [1] Salimans, Tim, and Richard Chen. "Learning montezuma's revenge from a single demonstration." arXiv preprint arXiv:1812.03381 (2018).

---

> > > ### Author Response · Authors · 2022-06-21
> > > **Clarifications to Reviewer 3Jg7**
> > >
> > > Thank you for your input.
> > >
> > > We need to make it clear that we are proposing a novel Self-Supervised Imitation Learning approach, not a novel RL method. Therefore we are working with the two following assumptions:
> > >
> > > 1) We have observations from an expert performing the actions in a given environment (the so-called "optimal policy"), often with a goal in mind (though that is not a hard constraint). In our context, an _optimal policy_ is an unknown function performed by the expert, and its optimality is not necessarily measurable by a known reward function.
> > >
> > > 2) The observations from the expert are not annotated, i.e., we do not have access to the actions of the expert in a given time, but only to its trajectory.
> > >
> > > Our main objective in this work is to **increase** **the** **efficiency** **of** **the** **Imitation** **Learning** **agent**, i.e., boost its results even when very few observations from the expert are available. We design a method that combines the general idea of Imitation Learning from Observations (though with several modifications in order to make it more effective) and that includes an inner RL optimization procedure to increase the agent's efficiency.
> > >
> > > In our experiments, we show that our proposed method (RIL) performs better (and more efficiently) than all Imitation Learning approaches that **share** **the** **same** **assumptions** **than** **RIL** (observations from an expert without annotations). For _completeness_, we also show that it performs better than Deep RL methods, though that is not necessarily a fair comparison since Deep RL methods do not have access to the expert trajectories. Our point here, however, is to show that we can achieve a good trade-off in terms of efficiency and effectiveness by designing a self-supervised IL method that leverages from the kind of environment exploration that RL provides. Indeed, we show in the ablation studies that RIL benefits from **a** **single** expert trajectory, making it much more efficient than the baselines.
> > >
> > > We are executing experiments in the suggested Atari datasets, but we also want to make it clear that our goal is not to outperform the state-of-the-art RL methods in those environments, but to be as effective (or perhaps more effective?) than an expert agent that acts in the environment while using as little observations (ideally a single trajectory) from the expert as possible.

---

### Review · Reviewer_T2Ws · 2022-06-15

**Summary Of Contributions:**

The authors propose to mix Imitation Learning and Reinforcement Learning. On their journey, they position their approach with respect to related work, propose an algorithm RIL, enounce experimental hypotheses, perform a large variety of experiments, and present the empirical results.

**Broader Impact Concerns:**

My little understanding of what/why/how things are made in this submission did not allow me to formulate any broader impact concerns analysis.

**Requested Changes:**

This section enumerates remarks and questions in an unorganized manner (in chronological order):
  - Abstract: *it requires a lot of computation* => I'm not sure what it means given that the RIL algorithm has an inner RL optimization loop.
  - Abstract: *a method that learns an optimal policy* => RIL looks like IL with some additional RL components, but if the objective is to learn an optimal policy, then it is more like IL-guided RL. **Nowhere in the submission, the objective of the RIL algorithm is clearly stated.**
  - Introduction 2nd paragraph: *at any given time* => no, a policy defines what the agent does (not should do) in any context. It is a function from the state space to a distribution over actions.
  - Introduction 2nd paragraph: *Value-based methods compute the optimal policy* => no, they search for the optimal policy.
  - Introduction 2nd paragraph: *The assumption that the only information available to the learning agent are the immediate environmental rewards* => the information signal is much richer than that with the transition function. Yann Le Cun even made this (in)famous quote that the reward is only the cherry on the machine learning cake.
  - Figure 1: the figure is absolutely impossible to understand when it is introduced. Not a single notation is introduced, the figure is not explained, etc.
  - Introduction 4th paragraph: *both performance and average episodic reward (AER) metrics*. These are not introduced and their difference is unclear at this point. We learn later that performance is actually just a rescaling of AER so they are simply the same indicator presented in a different way.
  - Related work: RL related work is too much centered on deep RL literature, it is also outdated, and off-topic (not connected to RIL).
  - Related work: *BC [...] learns how to approximate the agent’s trajectory from the expert’s* => not really. BC intends to reproduce the policy by predicting the action as a function of the state.
  - Related work: *require the expert policy or ground-truth labels to provide feedback to the agent.* => it was very unclear to me what meant ground-truth label is the RL setting. I understood much later that it meant the action performed by the expert. Is it really a strong assumption that the expert actions are observable?
  - Problem formulation: *Solving an MDP yields a stochastic policy π(a | s) with a probability distribution over actions for an agent in state s to perform* => I have no idea what this sentence intends to say. Solving an MDP means to find a policy optimizing an objective function.
  - Self-supervised Imitation Learning: *the policy (π_ϕ) acts as a stationary model predicting the most likely action a given s_t.* => a policy is a function of the state that returns a distribution over actions. It is not predicting anything. If the authors were referring to the Imitation policy, then it intends to reproduce not the most likely action, but the state-conditioned distribution of actions of the behavioral policy.
  - Self-supervised Imitation Learning: *balances I^s* => at this point, we don't know what I^s is and what balance is being looked after.
  - Self-supervised Imitation Learning: *that reach the environment goal* => does the environment need to be goal-based?
  - Self-supervised Imitation Learning: *This process allows the model to maintain a weighted distribution between the random and updated policy samples and avoid local minima since the probability of actions vanishing in each iteration is minimal.* I cannot understand this sentence. What weights are we talking about? What local minima? With respect to which objective? And finally, what does it mean to have the probability of action to vanish to be minimal? Minimal inside which set?
  - Exploration with Neural Networks: how is ε-greedy exploration with neural networks?
  - Exploration with Neural Networks: *to approximate the optimal Q-function in Equation 1* => there is no optimal q-function in equation 1, only a q-learning update.
  - Combining IL and RL: *(iv)* Is T^e different from dataset D?
  - Combining IL and RL: *(vi) use a sampling mechanism* => is it detailed somewhere? Is it eq 2?
  - Algorithm 1 lines 10-13: What is this set of environments? What do you mean by π_φ to solve environment e? What does it mean that π_φ is in turn updated until convergence to different objectives?
 - Combining IL and RL: *we compute the exploration ratio from the policy during the self-supervised learning component (Line 9)* => what is an exploration ratio? How is it computed. There are too much vagueness in many of the steps, **I cannot understand why it is done this way, and how to reproduce.**
  - Combining IL and RL: *covariance shift* => covariate shift.
  - Combining IL and RL: *k the slope of curvature* => it is another hyperparameter?
  - Equation 2: what is a lim sup of a dataset? The text explains well what it means: it is the number of samples coming from I^pos but the formula return a number that is smaller than 1. Should I understand that it is a ratio of samples coming from I^pos?
  - Equation 2: We may also notice that this formula is equivalent to $\frac{1}{1+(\frac{n}{e}-1)^k}$. This formula is weird (and again not motivated or even explained) since when e tends to $\infty$, it changes the sign inside the parenthesis, yielding non-monotonic behaviors.
  - Experimental methodology: *and its latest version (Schaul et al., 2015) (DQN2)* => it is not the *latest version* of DQN (whatever is means).
  - Policy optimization behavior: *IL and RL methods work on the same premise that an agent needs to learn an approximation to a theoretical
optimal policy in the form of an MDP.* Do you mean that both approaches return a policy? Because RL does not aim at learning a policy, but at yielding maximum discounted sum of rewards. In contrast, IL indeed aims at reproducing a policy, which may be interpreted in different ways (minimize policy KL or state density KL to name two).
  - Policy optimization behavior: *First, we compare all models with π∗ and compute the difference with the probability of all possible actions.* => A divergence is not a difference.
  - Policy optimization behavior: *the KL Divergence using one-hot encodings for the specific action given a state* => I don't understand what it means. Which specific action? Formalizing with actual equations would help to understand the difference between KL-divergence and KL-divergence*.
  - Policy optimization behavior: *When comparing π∗ to the other policies, we can see that in the middle of the valley...* => I do not understand what in Figure 2, we are asked to look at (Figure 2 is composed of 4 sub-figures).
  - Figure 2: what is a maximum a posteriori probability?
  - Figure 2a (and the others): why do probabilities have negative values?
  - Table 1: how can RIL yield more reward than the optimal policy? It raises the question: with respect to what is it optimal?
  - Table 1: it is suspicious that both RL and IL get exactly the same KL divergence*. Maybe it's possible. I can't tell since I did not understand how it is computed.
  - Policy optimization behavior: *Figure 3 illustrates how close RIL is from π∗* => why would we look at this? IL has for objective to reproduce another policy, and RL for objective to maximize return, which is only distantly connected to mimicking a specific policy (there may be several RL (near-)optimal policies).

After page 8, I felt that I was too lost to continue the reading and be able to provide relevant feedback on the paper. I will only mention that I've noticed that RIL consistently obtains performance strictly higher than 1, although 1 is supposed to be the performance of the optimal policy, once again raising the question of what is the optimal policy.

**Strengths And Weaknesses:**

Strengths: the scientific methodology is complete: positioning, algorithmic innovation, empirical hypothesis, and empirical validation. The figures are high quality.

Weakness: unfortunately, the writing and formalization are unclear. In my opinion, it undermines too much the submission to recommend acceptance. After reading the paper, I am left not understanding the motivation, the objectives, the claims, the empirical results, etc. As a result, I am unable to answer positively to the main reviewing question: *Are the claims made in the submission supported by accurate, convincing and clear evidence?* Next section provides many examples of imprecise formulations and inconsistent claims.

---

> ### Author Response · Authors · 2022-06-21
> **Response to Reviewer T2Ws**
>
> Thank you for your input on our work.
> We have read your points and agree that we should rework part of the first section to clarify the objective of our work and correct some of the theoretical elements, and we will change all points listed by you in the final version of this work.
>
> To be very clear, we are proposing a novel Self-Supervised Imitation Learning approach, not a novel RL method. Therefore we are working with the two following assumptions:
>
> 1) We have observations from an expert performing the actions in a given environment (the so-called "optimal policy"), often with a goal in mind (though that is not a hard constraint). In our context, an _optimal policy_ is an unknown function performed by the expert, and its optimality is not necessarily measurable by a known reward function.
>
> 2) The observations from the expert are not annotated, i.e., we do not have access to the actions of the expert in a given time, but only to its trajectory.
>
> Our main objective in this work is to **increase the efficiency of the Imitation Learning agent**, i.e., boost its results even when very few observations from the expert are available. We design a method that combines the general idea of Imitation Learning from Observations (though with several modifications in order to make it more effective) and that includes an inner RL optimization procedure to increase the agent's efficiency.
>
> In our experiments, we show that our proposed method (RIL) performs better (and more efficiently) than all Imitation Learning approaches that **share the same assumptions than RIL** (observations from an expert without annotations). For _completeness_, we also show that it performs better than Deep RL methods, though that is not necessarily a fair comparison since Deep RL methods do not have access to the expert trajectories. Our point here, however, is to show that we can achieve a good trade-off in terms of efficiency and effectiveness by designing a self-supervised IL method that leverages from the kind of environment exploration that RL provides. Indeed, we show in the ablation studies that RIL benefits from **a single** expert trajectory, making it much more efficient than the baselines.
>
> Below are answer your specific comments:
>
> 1) **Self-supervised Imitation Learning: the policy ($\pi_\phi$) acts as a stationary model predicting the most likely action a given $s_t$. => a policy is a function of the state that returns a distribution over actions. It is not predicting anything. If the authors were referring to the Imitation policy, then it intends to reproduce not the most likely action, but the state-conditioned distribution of actions of the behavioral policy.**
>
> Here we are referencing the second sub-model of RIL, which acts as a policy model but also has the exploration mechanism mentioned in our work. We understand the confusion regarding what the policy is and what the method does outside the policy and we will update our work to better reflect what we are talking about.
>
> 2) **Self-supervised Imitation Learning: balances $I^s$ => at this point, we don't know what $I^s$ is and what balance is being looked after.**
>
> You are correct, there were no previous mention on what $I^s$ is. $I^s$ in the IL context is the current samples used in the Inverse Dynamics Model. It is made out of $I^{pre}$, samples from the random policy, and $I^{pos}$, samples from the current learned policy. The balance we are trying to achieve here is to have samples that are closer to the expert than the random policy were.
>
> 3) **Self-supervised Imitation Learning: that reach the environment goal => does the environment need to be goal-based?**
>
> There needs to be a goal previously defined for the algorithm. This goal does not have to be the same goal from the environment itself. For example, when using the MountainCar environment, the agent does not need to reach the flag with little to no reward, but only reach the flag. Another example is the LunarLander env, where the agent could perhaps only land the ship anywhere, instead of in-between the flags goal.
>
> Continues in the next comment.

---

> > ### Author Response · Authors · 2022-06-21
> > **Comments 4 to 7**
> >
> > 4) **Self-supervised Imitation Learning: This process allows the model to maintain a weighted distribution between the random and updated policy samples and avoid local minima since the probability of actions vanishing in each iteration is minimal. I cannot understand this sentence. What weights are we talking about? What local minima? With respect to which objective? And finally, what does it mean to have the probability of action to vanish to be minimal? Minimal inside which set?**
> >
> > Here in this sentence we refer to $I^s$. In [2], the authors discovered that weighting the amount of samples from the expert into $I^s$ was better than using the approach from [1]. With that sampling mechanism, the iterative process does not have cases where an action would vanish (be very small) from the action distribution over the policy. That causes problems on more complex environments where the agent would never predict an action (no matter from what state) and be stuck on local minima (doing a loop between suboptimal actions - going back and forth in a maze for example). Finally, the probability of action vanishing to be minimal refers to the scenario where the model does not stop annotating actions for the expert samples.
> >
> >
> > 5) **Exploration with Neural Networks: how is $\epsilon$-greedy exploration with neural networks?**
> >
> > The $\epsilon$-greedy mechanism with neural networks works by assigning an initial value for $\epsilon$, and this value decays with time. When the number is lower than $\epsilon$, the method does not use the distribution from the policy: it samples from a random distribution. With time this value will decay to zero, and thus the method will only use the distribution from the policy to sample the action (usually an argmax from the softmax distribution).
> >
> > 6) **Combining IL and RL: (iv) Is $T^e$ different from dataset D?**
> >
> > No, you are correct. We should have used $D$ instead of $T^e$.
> >
> > 7) **Combining IL and RL: (vi) use a sampling mechanism => is it detailed somewhere? Is it eq 2?**
> >
> > You are right, we forgot to insert that in the paper. For obtaining the samples from post-demonstrations, we first select the distribution of actions given an environment and the current policy $P(A|E;\mathcal{I}^{pos})$. It considers only successful runs from $\mathcal{I}^{pos}$, i.e., only state-action sequences in which the agent was able to achieve the environment goal. We represent it as $v_e$ in: $P(A|E;\mathcal{I}^{pos}) = \frac{\sum_{e=1}^{|E|} v_e \times  P(A|e)}{|E|}$, where $v_e$ is set to 1 if the agent achieves the environment goal or zero otherwise, and $E$ is the set of runs in an environment. The only assumption we make of the available demonstration data is that most image sequences are those of the demonstrator achieving a goal, even if such demonstrations have no label indicating whether the demonstration achieves the goal or not. The intuition for using the post-demonstrations from successful runs alone is that if a policy is not able to achieve the environment goal, then the post-demonstration alone is not enough to close the gap between what the model previously learned with $\mathcal{I}^{pre}$ and what the expert performs within the environment.
> > Using only successful runs also gives us a more accurate distribution of the expert's actions since we are only using those observations that achieved the objective instead of the random performance that maps the dynamics of the agent within a balanced dataset of actions.
> > By not adding unsuccessful runs to the training dataset, we solve the problem in which the model in [1] degrades the performance in both models.  With the distribution of actions for successful runs, we select all samples $\mathcal{I}^{pos}$ from those runs, assuming each action follows the distribution of the equation.
> >
> > The next step is to sample from the pre-demonstrations $\mathcal{I}^{pre}$ with the inverse probability of the post-demonstrations, i.e., the loss probability distribution given by $1- P(A|E;\mathcal{I}^{pos})$. In a nutshell, the samples (observations) that comprise the novel post-demonstration dataset are sampled proportionally to $P(A|E;\mathcal{I}^{pos})$ for winning executions, and the dataset is filled with the pre-demonstrations proportional to the number of losses.
> > The final set $\mathcal{I}^{s}$ contains the concatenation of both $\mathcal{I}^{pre}$ and $\mathcal{I}^{pos}$ samples.
> >
> > Complementing the training dataset with random demonstrations has two main advantages. First, it helps the model to avoid overfitting the policy demonstrations. Second, in the early iterations, when the policy generates very few successful runs, and its action distribution is entirely dissimilar to the expert, the training data will guarantee improved exploration by the IDM. With the win-loss probability, we force the training data to be closer to the expert demonstration than to the random data, which boosts the model capability of imitating the expert.

---

> > > ### Author Response · Authors · 2022-06-21
> > > **Comments 8 to 12**
> > >
> > > 8) **Algorithm 1 lines 10-13: What is this set of environments? What do you mean by $\pi_\phi$ to solve environment e? What does it mean that $\pi_\phi$ is in turn updated until convergence to different objectives?**
> > >
> > > Here when we mention _setting an environment_ is regarding setting a seed (so the same initialization is not seen during training and validation). By "$\pi_\phi$ solving the environment" we meant using the current learned policy in the environment to gather samples. We understand that perhaps the current policy might not achieve the goal of the environment.
> > > Finally, we repeat the loop between IL and RL understanding that they do optimize different objectives, and set a threshold of what is considered an improvement. Most of what RIL does depends on it updating its policy weights by finding a middle-term between IL and RL. By using the expert as a guide RIL can optimize the distribution of the actions according to the expert and by using the information in the reward function it can correct itself.
> > >
> > > 9) **Combining IL and RL: we compute the exploration ratio from the policy during the self-supervised learning component (Line 9) => what is an exploration ratio? How is it computed. There are too much vagueness in many of the steps, I cannot understand why it is done this way, and how to reproduce.**
> > >
> > > The original behavioral cloning framework uses the _maximum a posteriori_ (MAP) estimation, i.e., it predicts the action with the greatest probability given by the model for a pair of states. By using MAP predictions, we identified several cases in which the model is still relatively unsure about the correct action, especially in earlier iterations, leading to undesired local minima.
> > > We borrow a simple solution from [2], which is to sample the actions from the softmax distribution of both models (IDM during the expert labeling and policy model during the execution of the environment).
> > > By not using the MAP estimation, we create a stochastic policy capable of further exploration in early iterations considering the model uncertainty. We show that creating those samples allows the IDM to converge in fewer iterations, since the sampling method guarantees a more sparse dataset consisting of $\mathcal{I}^{pre}$ and $\mathcal{I}^{pos}$, and the stochastic policy guarantees more exploration of the search space for properly achieving the environment goal. Furthermore, in dynamic environments, the stochastic policy contributes to reaching the goal when deterministic behavior would not. This difference is vital for avoiding local minima during iterations. If the model were not capable of sampling a sub-optimal action during its training phase, the agent actions would resume for the most common action in the expert samples. By sampling the most frequent action, the policy is susceptible to looping between states, e.g., choosing left and right interchangeably.
> > >
> > > 10) **Combining IL and RL: k the slope of curvature => it is another hyperparameter?**
> > >
> > > Yes, in Eq. 2 there are 2 different hyperparameters. $n$ is the number of epochs we want to train RIL and $k$ being how the limit should behave, where $k = 1$ will result in a linear superior limit, and $k = 4$ will result in a sigmoid-like function with its inflection point being half of $n$.
> > >
> > > 11) **Equation 2: what is a lim sup of a dataset? The text explains well what it means: it is the number of samples coming from $I^pos$ but the formula return a number that is smaller than 1. Should I understand that it is a ratio of samples coming from $I^{pos}$?**
> > >
> > > Yes, you are correct, Eq. 2 established a ratio for sampling from $I^{pos}$. This is due to the covariate shifts that happens in [1] and [2]. By setting a more adequately distribution over time, the IDM model will not collapse or update its weights to heavily.
> > >
> > > 12) **Policy optimization behavior: IL and RL methods work on the same premise that an agent needs to learn an approximation to a theoretical optimal policy in the form of an MDP. Do you mean that both approaches return a policy? Because RL does not aim at learning a policy, but at yielding maximum discounted sum of rewards. In contrast, IL indeed aims at reproducing a policy, which may be interpreted in different ways (minimize policy KL or state density KL to name two).**
> > >
> > > Yes, what we had in mind is that both RL and IL will result in a policy, even though both approaches do not share the same objective (as mentioned in your comment). This is the reason for our first experiment where we compare RIL with a IL and RL method.
> > > We wanted to understand how RIL will compare in its decision making process when compared to policies that were created with different objectives.

---

> > > > ### Author Response · Authors · 2022-06-21
> > > > **Comments 13 to 18**
> > > >
> > > > 13) **Policy optimization behavior: the KL Divergence using one-hot encodings for the specific action given a state => I don't understand what it means. Which specific action? Formalizing with actual equations would help to understand the difference between KL-divergence and KL-divergence$*$.**
> > > >
> > > > We will include an equation in the final version. What we did was to set the argmax action to 1 and the remaining we set to zero.
> > > >
> > > > 14) **Figure 2: what is a maximum a posteriori probability?**
> > > >
> > > > Answered in question 9).
> > > >
> > > > 15) **Figure 2a (and the others): why do probabilities have negative values?**
> > > >
> > > > The values displayed in the figure are not the probabilites of the actions, but the value predicted by the model for the MAP action for $\pi^*$. Therefore, if $\pi^*$ MAP is action $1$, the other figures will present the same output value for that action in that model. We did not use a softmax function before presenting the values: we are presenting logits.
> > > >
> > > > 16) **Table 1: how can RIL yield more reward than the optimal policy? It raises the question: with respect to what is it optimal?**
> > > >
> > > > The IL literature defines the Expert as a hypothetical, optimal policy. So even though there may be better policies (in terms of reward), in the IL context the expert is an optimal policy as well. So for CartPole, MountainCar, and LunarLander we trained some RL agents with the solving criteria ($195$, $-110$, and $200$, respectively) as the threshold for an optimal policy.
> > > > For the Acrobot environment, we used the best agent we could train ($-85$). Therefore, during the experimental analysis for the "Policy Optimization Behavior" section, we did not want to create the best policy for the environment, only a policy that would fit the premise of optimality for IL. We will rename "optimal policy" to something along the lines of "solving policy", i.e., the agent meets the requirement for solving the environment according to [3].
> > > >
> > > >
> > > > 17) **Policy optimization behavior: Figure 3 illustrates how close RIL is from $\pi$$*$ => why would we look at this? IL has for objective to reproduce another policy, and RL for objective to maximize return, which is only distantly connected to mimicking a specific policy (there may be several RL (near-)optimal policies).**
> > > >
> > > > Our main research question in this work was _when combining RL and IL, how will the policy behave when compared to another solving agent? And how close will this behavior be to other IL and RL methods?_ We understand that IL tries to mimic the expert policy, while RL will optimize the sum of the reward function. We hypothesize that since RIL is optimizing both objectives iteratively, it could be a mashup from RL and IL, but never better than them separately. The experiment shows that when compared to the expert, RIL achieves a similar trajectory but not the same distribution. Thus, the KL-Divergence is the farthest, but it is the closest when considering the argmax values (the action taken when considering a greedy agent will be similar).
> > > > The experiment also shows that by combining RL and IL, a method can achieve rewards that are better than the trajectory used for guidance, which helps the model behave differently from the expert and closer to the RL agent.
> > > >
> > > > 18) **Regarding the experimental methodology.**
> > > >
> > > > We are working on experimenting with the Atari 2600 envs, so new results should follow soon.
> > > > We will post here as soon as we have them.
> > > >
> > > > [1] Behavioral Cloning from Observation - Torabi et al.
> > > >
> > > > [2] Imitating Unknown Policies via Exploration - Gavenski et al.
> > > >
> > > > [3] Efficient memory-based learning for robot control - Moore, Andrew William

---

> > > > > ### Comment · Reviewer_T2Ws · 2022-06-28
> > > > > **Response to response**
> > > > >
> > > > > Thanks for the extensive response.
> > > > >
> > > > > I deeply appreciate the responses to my comments, their extensivity, their honesty, and even their overall clarity. However, the changes I asked for are too broad for me to have a clear picture of the quality of the corrected paper. As a consequence, I can only recommend rejection.
> > > > >
> > > > > NB: If the objective is IL, why are the experiments evaluated with respect to the policy returns?

---

### Review · Reviewer_m4Pq · 2022-06-15

**Summary Of Contributions:**

The authors propose an algorithm that tackles the Learning from Demonstration problem in the more specific setting more the demonstrations contain only observations and no action (Learning from Observations).
The agent thus has access to the environment to interact with, to the reward information and to demonstrations with only observations in it.
The authors propose to:
- initialize a policy $\pi_{\theta}$
- learn an inverse model to predict $a_t$ given $s_t$ and $s_{t+1}$ based on random exploration
- train $\pi_{\theta}$ with behavioral cloning loss on the demonstration dataset with the actions predicted by the inverse model.
- interact with the environment to
     - train $\pi_{\theta}$ with TD-loss on the reward
     - augment the dataset the inverse model is trained on

There is a goal sampling technique mentioned in Algorithm 1 that I don’t understand given that all environments considered are not goal-conditioned.


**Broader Impact Concerns:**

No particular concern.

**Requested Changes:**

- A much clearer introduction of the notations
- A clear statement of the contributions
- A refactoring of section 3 which completely mix contributions with related work
- A clearer description of the algorithm.
- Proper citations of the related work when using techniques that were previously used
- Mathematical expression when needed (as example when introducing the considered KL-divergence).
- A clear explanation of the baselines.
- A choice of stronger baselines.
- A clear explanation of the hyperparameter selection process with a single set for all environments.
- At least one higher dimensional environment


**Strengths And Weaknesses:**

# Strength
It is an interesting setup to tackle LfD from observation only.

# Weaknesses
- Writing
  - It was not clear for me until page 5 what was the actual setup considered (and I am still not a 100% sure). To understand what is the tackled problem (LfD from obs.), the reader has to gather information from 6 different places. Some details about it are never even mentioned, like the restriction to the discrete action setup.
  - The algorithm explanation lacks a lot of details. E.g. The authors write “using $\pi_{\theta}$ as $I^s$” although $I^s$ has not been introduced before. The authors probably rely on figure 1 for the reader to understand on its own.
  - A lot of the notations used are actually never introduced, or change across the explanation (r vs R, $\pi$ vs $\pi_{\theta}$ etc…).
- Algorithm and contributions
  - The contributions are not clearly stated. The algorithm is described as a bag of tricks that are poorly linked to the relevant literature.
  - Using DQN with a BC loss is actually the core of the DQfD [1] algorithm (that was extended in DDPGfD[2] and R2D3 [3], or used differently in AQuaDem [4]). Inverse models have been extensively studied in the literature, e.g. for reward bonus like in ICM [5] but in many more.
  - Tricks are stated as contributions like acting following softmax(logits) but various uncited work have been doing this in the past years, like SoftDQN [6] or Munchausen RL [7].
  - The authors don’t bring the reader any intuition or mathematical explanation why the algorithm should work.

- Experimental setup
  - The experimental setup is weak in the following sense: many crucial details are lacking: where do the demonstrations come from? How many steps the algorithms are trained on? Proper description of the baselines: baseline scores are very low on these very simple environments and I strongly suspect the use of weak baselines. GAIL has been improved a lot since initial publication, as shown in the following work [8]
  - Authors consider only very low dimensional environments where learning the model is probably very simple. At least one larger example would really be welcome to showcase the performance of the algorithm.
  - An improper hyperparameter selection method where different HPs are needed in every environment.

[1] https://arxiv.org/abs/1704.03732
[2] https://arxiv.org/pdf/1707.08817.pdf
[3] https://arxiv.org/abs/1909.01387
[4] https://arxiv.org/abs/2110.10149
[5] https://arxiv.org/pdf/1705.05363.pdf
[6] https://arxiv.org/abs/1912.10891
[7] https://arxiv.org/abs/2007.14430
[8] https://arxiv.org/abs/2106.00672

---

> ### Author Response · Authors · 2022-06-21
> **Response to Reviewer m4Pq**
>
> Thank you for your input.
>
> We need to make it clear that we are proposing a novel Self-Supervised Imitation Learning approach, not a novel RL method. Therefore we are working with the two following assumptions:
>
> 1) We have observations from an expert performing the actions in a given environment (the so-called "optimal policy"), often with a goal in mind (though that is not a hard constraint). In our context, an _optimal policy_ is an unknown function performed by the expert, and its optimality is not necessarily measurable by a known reward function.
>
> 2) The observations from the expert are not annotated, i.e., we do not have access to the actions of the expert in a given time, but only to its trajectory.
>
> Our main objective in this work is to **increase the efficiency of the Imitation Learning agent** , i.e., boost its results even when very few observations from the expert are available. We design a method that combines the general idea of Imitation Learning from Observations (though with several modifications in order to make it more effective) and that includes an inner RL optimization procedure to increase the agent's efficiency.
>
> In our experiments, we show that our proposed method (RIL) performs better (and more efficiently) than all Imitation Learning approaches that **share the same assumptions than RIL** (observations from an expert without annotations). For _completeness_, we also show that it performs better than Deep RL methods, though that is not necessarily a fair comparison since Deep RL methods do not have access to the expert trajectories. Our point here, however, is to show that we can achieve a good trade-off in terms of efficiency and effectiveness by designing a self-supervised IL method that leverages from the kind of environment exploration that RL provides. Indeed, we show in the ablation studies that RIL benefits from **a single** expert trajectory, making it much more efficient than the baselines.
>
> Specific notes below:
>
> 1) **Writing and notation.**
>
> This work uses notions that are part of the IL literature and perhaps should have been better explained here to avoid further confusion (we will introduce it in our rewriting of the mentioned sections).
>
> 2) **Algorithms and Contributions.**
>
> We take note of the suggested literature and will be sure to add them to our references. In this work, we use a more robust IL approach that works self-supervisedly, which does not depend on finding expert samples with labels. RIL uses only random and self-generated samples to learn a state demonstration, and [1] also showed that using this iterative process with no ground-truth labels helps achieving better results. We will be sure to add the softmax/logits literature, though we have mentioned [2], which uses it as an exploration mechanism as we do.
>
> 3) **Experimental setup and methodology.**
>
> We are working on using the Atari benchmark to showcase a more complex scenario for the experimental setup.
> Specific questions on the methodology:
>
> 3.1) **Where do the demonstrations come from?**
>
> The demonstrations are from different RL agents trained using the Stable Baselines framework. We will add which approach was used for each environment in the final version of the paper.
>
> 3.2) **How many steps the algorithms are trained on?**
>
> Since RIL is an interactive process that interleaves IL and RL, we train them in epochs. An epoch is considered a forward pass in the expert dataset (size may vary, and this information is displayed in Table 6) and $100$ consecutive runs for the RL inner optimization. Table 3 presents the number of timesteps taken before reaching the maximum reward. For that, we compute the number of expert samples plus the number of time steps used in the RL optimization.
>
>
> [1] Behavioral Cloning from Observation - Torabi et al.
>
> [2] Imitating Unknown Policies via Exploration - Gavenski et al.

---

> > ### Comment · Reviewer_m4Pq · 2022-06-28
> > **Answer to reviewers**
> >
> > Thanks for your answer.
> >
> > Overall I think the motivation is still not clear: "increase the efficiency of the Imitation Learning agent" -> I do not agree with the authors. You use the reward function, it is not imitation learning. This kind of misunderstanding is really present in the paper and require quite deep corrections.
> >
> > I read the author's rebuttal but I still think the changes required for publication are too important to recommend acceptance.

---

### Decision · Action_Editors · 2022-07-25

**Recommendation:** Reject

**Comment:**

There has been extensive discussions between the reviewers and the authors. The reviewers comments more or less highlight similar strength and weaknesses. They appreciated the original work and the fact that this line of work is very relevant to the journal as well as the timeliness. It brings a new perspective on relevant problems in the imitation / RL community.

Yet they also unanimously agree on the lack of clarity of the paper and major discrepancies between claims and experiments (like using the reward in an imitation learning setting). This is the main criteria for acceptance at TMLR and these issues unfortunately prevent publication at this stage. This is why all reviewers recommended to reject the paper this time.

The authors seem to have understood most of the issues from the reviewers comments. I really appreciated the discussion and I'm sure it will help the authors coming up with a much better version of their paper. Yet, at this moment, there is too much work to be done for relying on this discussion so as to accept the paper.